# Application Strategies of Super-Enhancer RNA in Cardiovascular Diseases

**DOI:** 10.3390/biomedicines13010117

**Published:** 2025-01-07

**Authors:** Yi He, Yuwei Cai, Yanyan Cao, Yan Wang, Jing Wang, Hu Ding

**Affiliations:** 1Division of Cardiology, Departments of Internal Medicine, Tongji Hospital, Tongji Medical College, Huazhong University of Science and Technology, Wuhan 430030, China; m202276202@hust.edu.cn (Y.H.); cyw12621@163.com (Y.C.); cyy1016@163.com (Y.C.); newswangyan@tjh.tjmu.edu.cn (Y.W.); 2Hubei Key Laboratory of Genetics and Molecular Mechanisms of Cardiological Disorders, Wuhan 430030, China; 3Genetic Diagnosis Center, Tongji Hospital, Tongji Medical College, Huazhong University of Science and Technology, Wuhan 430030, China; 4Key Laboratory of Vascular Aging, Ministry of Education, Tongji Hospital, Tongji Medical College, Huazhong University of Science and Technology, Wuhan 430030, China

**Keywords:** cardiovascular diseases (CVDs), enhancer RNA (eRNA), super-enhancer RNA (seRNA)

## Abstract

Cardiovascular diseases (CVDs) are a leading cause of death worldwide, and new therapeutic strategies are urgently needed. In recent years, enhancer RNAs (eRNAs) have gradually attracted attention because they offer new directions for the treatment of CVDs. Super-enhancer RNAs (seRNAs) are a subset of non-coding RNAs that are transcribed from regions of the genome known as super enhancers, which are large clusters of enhancers with a high density of transcription factors and cofactors. These regions play a pivotal role in regulating genes involved in cell identity and disease progression. This article reviews the characteristics of seRNAs, their expression patterns, and regulatory mechanisms in the cardiovascular system. We also explore their role in the occurrence and development of CVDs, as well as their potential as diagnostic biomarkers and therapeutic targets. Currently, therapies targeting seRNAs are a research hotspot. The development of specific inhibitors or activators is expected to facilitate precise interventions for CVDs. In addition, the use of gene editing techniques to modify relevant eRNA introduces new possibilities for disease treatment. This review aims to provide a comprehensive overview of seRNAs in CVDs and discusses their potential as a novel class of therapeutic targets.

## 1. Introduction

Cardiovascular diseases (CVDs) are becoming increasingly prevalent worldwide. With an aging population and widespread adoption of unhealthy lifestyles, the incidence of CVDs continues to increase, affecting a growing number of younger individuals. As the leading cause of death worldwide, CVDs impose a substantial economic burden on families and society [1,2,3]. The treatment of CVD remains a significant challenge in the medical field. Although traditional therapeutic approaches can alleviate symptoms and slow disease progression, they have several limitations. In addition, the pathogenesis of CVD is complex and involves abnormalities in multiple genes and signaling pathways, making the selection of therapeutic targets particularly challenging. Therefore, it is important to explore new biomarkers and therapeutic targets as early as possible for their diagnosis and treatment of CVDs.

With the rapid development of high-throughput sequencing technology, many non-coding RNAs (ncRNAs) have been shown to play crucial physiological and pathophysiological roles in CVDs [4,5]. ncRNAs are a class of RNA molecules that do not encode proteins, but instead, they regulate a wide range of cellular processes through intricate mechanisms (Table 1). These molecules are broadly classified into two major categories based on their length: small non-coding RNAs (sncRNAs) and long non-coding RNAs (lncRNAs). Long non-coding RNAs, which are typically greater than 200 nucleotides (nt) in length, have been shown to regulate gene expression at various levels, including chromatin remodeling, transcriptional regulation, and post-transcriptional processing [6]. Some well-known lncRNAs, such as lncRNAs MALAT1 [6,7,8], H19 [9,10,11,12], HOTAIR [13,14,15], and MEG3 [16,17,18], have been shown to play significant roles in CVDs by modulating processes like inflammation, fibrosis, and vascular remodeling. Small non-coding RNAs include microRNAs (miRNAs), piwi-interacting RNAs (piRNAs), tsRNAs (tRNA-derived small RNAs), etc., which are also known for their roles in post-transcriptional regulation [19]. With recent breakthroughs in molecular biology and genetics, a novel class of molecules known as super-enhancer RNAs (seRNAs) has attracted significant attention in cardiovascular research [20,21]. The discovery of seRNAs has provided new opportunities to advance our understanding and treatment of CVDs.

In April 2013, Young published a pioneering study in Cell in which he first introduced SEs [22]. Super-enhancers (SEs) are large, specialized genomic regulatory elements that play critical roles in the regulation of gene expression [21,23]. Typically, they consist of multiple clustered enhancers spanning 8–10 kb or larger genomic regions. Compared to typical enhancers (TEs), SEs exhibit the following characteristics (Figure 1): (1) Higher transcriptional activity. SEs are rich in binding sites that recruit a variety of transcription factors (TFs), RNA polymerase II, and co-activators that drive elevated transcription levels of associated genes. This leads to robust gene expression in cells with active SEs. (2) Distinct epigenetic modifications. SEs are marked by specific histone modifications, such as H3K4me1 (histone H3 lysine 4 monomethylation) and H3K27ac (histone H3 lysine 27 acetylation), and the presence of chromatin regulators, such as p300 (a transcriptional coactivator) and BET family proteins (bromodomain and extra-terminal domain proteins, e.g., BRD4 (Bromodomain-containing protein 4), a group of chromatin readers that bind to acetylated lysine residues, such as H3K27ac). These indicate open chromatin regions that are occupied by a range of auxiliary factors such as mediators and cohesins, which further facilitate transcriptional activation. (3) Increased DNase I hypersensitivity. SEs are highly sensitive to DNase I digestion, suggesting an open, accessible chromatin structure that is readily available for TF binding and other regulatory activities. (4) SEs have a stronger and more focused effect; therefore, when the genes they regulate are disrupted, their impact on the body is usually more severe. This is especially true for CVDs, which depend on the expression of specific genes. A decrease in the expression of SE-regulated genes often leads to more serious health issues [22,24]. (5) SEs are associated with a high density of RNA polymerase II, which enables the transcription of large amounts of SE-associated RNA [25,26,27].

In 2010, Greenberg et al. made the groundbreaking discovery of enhancer RNAs (eRNAs), non-coding RNA molecules transcribed from enhancer regions, which rapidly became a hot topic in gene regulation research [28]. eRNAs play a key role in regulating gene expression, particularly in processes like cellular differentiation and identity. SEs represent a highly dynamic and specialized layer of genetic regulation. Their distinct chromatin features, high transcriptional activity, and involvement in the expression of key genes are important for cellular differentiation, identity, and disease mechanisms. seRNAs were once regarded as transcriptional noise without functional significance, resulting from spurious transcription in open chromatin regions [29]. The emerging discovery of super-enhancer RNAs (seRNAs) has further deepened our understanding of these critical regulatory elements. Non-coding RNAs transcribed from or interacting with SE regions are referred to as super-enhancer RNAs (seRNAs) [30,31]. Recent investigations into seRNAs and their involvement in pathological states have provided important insights into disease mechanisms [32]. For example, in cancer, seRNAs, such as UCA1 [33], HCCL5 [34], LNC01503 [35], and CCAT1 [36], play a crucial role in tumorigenesis by mediating oncogenic signaling pathways, which contribute to processes including cell proliferation, apoptosis, epithelial–mesenchymal transition (EMT), extracellular matrix remodeling, and angiogenesis [32]. In autoimmune diseases, the seRNA IFNG-R-49 has been implicated in the pathogenesis of inflammatory bowel disease (IBD) [37]. The study of SEs and their associated RNAs is an evolving field, with ongoing research revealing their contributions to various diseases, including CVDs [20,21,38,39].

Currently, research on the role of seRNAs in CVDs is limited, and most of their functions and mechanisms remain poorly understood. Therefore, investigating the expression and functional roles of seRNAs is a crucial area of study. Here, we reviewed the role of seRNAs in CVDs, focusing on seRNAs that are relevant to CVD pathogenesis. We begin by providing a brief overview of the mechanisms and physiological functions of seRNAs, followed by an exploration of the effect of their dysregulation on CVDs. Finally, we discuss the potential of non-coding seRNAs as diagnostic and prognostic biomarkers and their emerging role as targets for novel therapeutic strategies.
Figure 1The classification of typical enhancers (TEs) and super enhancers (SEs). (**a**) TEs (left) bind transcription factors to recruit coactivators (e.g., p300, mediators, BRD4) and regulate gene expression. In contrast, SEs (right) are large clusters of enhancers bound by master transcription factors, concentrating coactivators to drive strong expression of genes important for cell identity. This is enhanced by phase separation, driven by weak and multivalent interactions of disordered regions [40]. (**b**) The ranking number of enhancers based on the chromatin immunoprecipitation sequencing (ChIP-seq) signal of H3K27ac is used to distinguish between SEs and TEs (created in https://BioRender.com).
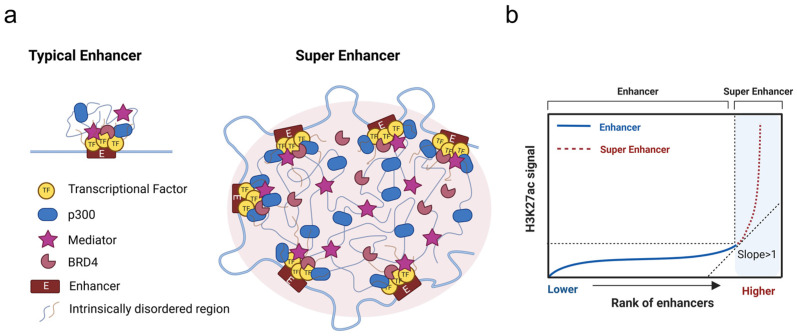

biomedicines-13-00117-t001_Table 1Table 1Classification of non-coding RNAs (ncRNAs).ClassTypeFeatures and FunctionExamplesLong Non-Coding RNAslincRNAs>200 nt, transcribed from regions located between protein-coding genes, involved in chromatin remodeling, transcription regulation, and modulation of gene expression. H19 [10], XIST [41],MALAT1 [7]Promoter-associated lncRNAsTranscribed from regions near or overlapping the promoters of genes. NEAT1 [42], HOTTIP [43]circRNAsCovalently closed non-linear RNAs formed by back-splicing of exons act as miRNA sponges or interact with RNA-binding proteins to regulate transcription and alternative splicing.CircRNA Galntl6 [44],CircRNA Samd4 [45]NATsTranscribed in the opposite direction of protein-coding genes, modulate gene expression through RNA-RNA interactions or chromatin modifications [46]. HOTAIR [15], MEG3 [16],ANRIL [47]eRNAs and seRNAsTranscribed from the enhancer region of a protein-coding gene, regulate enhancer-promoter interactions and facilitate transcriptional activation by modifying chromatin structure and recruitment of RNA polymerase.Bvht [48], CCAT-1 [36],Wisper [21],CARMEN [49]Small Non-Coding RNAsmiRNAs~22 nt, regulate gene expression post-transcriptionally by binding to 3′UTRs of target mRNAs, causing mRNA degradation or translation inhibition.miR-133 [50],miR-21 [51] piRNAs24–30 nt, interact with Piwi proteins, and are mainly involved in silencing transposons in germline cells and maintaining genome integrity.piR-823 [52], piRNA-30473 [53]tsRNAs15–45 nt, derived from tRNAs, regulate gene expression, stress response, cellular signaling, and protein translation.tRF-Gln-CTG-026 [54],tiRNA-Val-CAC-2 [55] YRNA and ysRNAs~70–100 nt, involved in regulating RNA stability, processing misfolded RNAs, and assembling RNA-protein complexes like the Ro ribonucleoprotein complex.s-RNYs [56] s-RNY1-5p [57]Other Non-Coding RNAssnRNAs~100–215 nt, key components of the spliceosome, involved in pre-mRNA splicing.U1, U2, U4, U7 [58] snoRNA~60–300 nt, guide RNA modifications (methylation or pseudouridylation) of rRNAs, involved in the maturation of rRNAs and ribosome biogenesis.SNORD15 [59]SRP RNA~300 nt, part of the Signal Recognition Particle, which guides ribosome–mRNA complexes to the endoplasmic reticulum for co-translational translocation. 7SL RNA [60]rRNAsStructural and catalytic components of ribosomes are essential for protein synthesis.18S,28S,5S,5.8S rRNAs [61]tRNAs~70–90 nt, involved in the translation process, transporting amino acids to ribosomes.tRNA-Ser, tRNA-Lys [62]lincRNAs: long intergenic non-coding RNAs; eRNAs: enhancer RNAs; seRNAs: super-enhancer RNAs; circRNAs: circular RNAs; NATs: natural antisense transcripts; miRNAs: microRNAs; piRNAs: piwi-interacting RNAs; tsRNAs: tRNA-derived small RNAs; ysRNAs: RNY (YRNA)-derived small RNAs; snRNAs: small nuclear RNAs; snoRNA: small nucleolar RNAs; SRP RNA: signal recognition particle RNA; rRNAs: ribosomal RNAs; tRNAs: transfer RNAs.


## 2. Methodology

Initially, we conducted a comprehensive literature search across several widely recognized scientific databases, including PubMed, Google Scholar, Scopus, and Web of Science. The search was performed using a combination of keywords such as “Super enhancer RNA”, “Super enhancer-associated RNA”, “Super enhancer”, “cardiovascular diseases (CVD)”, “heart disease”, “Cardiomyopathy”, “Hypertension”, “Atherosclerosis”, “Myocardial infarction”, “Heart failure”, and other relevant terms.

We included research studies that aligned with the theme of our review that discussed the role of seRNA in cardiovascular diseases and the potential mechanisms involved. In addition, relevant studies cited within these papers were also considered for inclusion. Each selected article was carefully analyzed and categorized based on its relevance to this review, with a particular focus on seRNA and its impact on CVD.

Finally, we synthesized the findings, identifying key patterns and mechanisms linking seRNA to cardiovascular pathogenesis.

## 3. Structure and Function of Super-Enhancer RNA

SEs have a highly ordered and compact spatial structure, forming a specific three-dimensional structure that facilitates interactions with other molecular entities to drive robust gene expression [63]. seRNAs are a class of non-coding RNAs strongly associated with transcriptional activation [64]. Unlike other lncRNAs, seRNAs are transcribed bidirectionally from SE regions that are highly modified by histone methylation and acetylation, resulting in high levels of histone H3K4me1, H3K4me2 (histone H3 lysine 4 dimethylation), and H3K27ac modifications [28,65]. In contrast to mRNAs, most seRNA sequences are relatively short (approximately 500 bp), non-polyadenylated, unstable, and predominantly localized in the nucleus [66,67]. A small subset of seRNAs transcribed from active enhancer-specific regions are longer (approximately 5 kb) and polyadenylated (Figure 2). Non-polyadenylated short seRNAs primarily exert cis-regulatory functions, whereas polyadenylated seRNAs in the nucleus, which are more stable, can participate in trans-regulatory roles.

seRNAs can be categorized into several types depending on their structure and function. Some seRNAs are long, contain polyadenylated tails enriched with specific nucleotide sequences, and form intricate secondary structures. Other seRNAs are relatively short, lack polyadenylated tails, and contain highly active transcriptional regions [68,69]. For instance, JUN-mediated seRNA-NPCM contains multiple loop structures that form an R-loop, which modulates the chromatin interaction between the SE and distal NDRG1 promoters, thus facilitating NPC translocation and enhancing its interaction with ACTA1 proteins [70]. In contrast, some seRNAs may adopt compact stem-loop structures that help stabilize their intracellular presence [71].

The generation of seRNAs involves a series of intricate molecular processes. Initially, the SE region undergoes chromatin opening, which exposes the DNA motifs to an active conformation. This conformation enables the recruitment of RNA polymerase II and other specific TFs to bind to particular gene regions, initiating the transcription of non-coding RNA [72] (Figure 2). Simultaneously, chromatin remodeling and histone modifications play pivotal roles in creating an environment conducive for transcription [73]. Regulatory factors that influence seRNA production include intracellular signaling pathways, environmental cues, and epigenetic modifications. For example, activation of certain cytokines can promote seRNA synthesis, whereas alterations in DNA methylation can suppress their expression.

seRNAs enhance transcriptional efficiency by forming complexes with specific TFs. In certain contexts, they can also recruit chromatin-remodeling complexes, altering the chromatin structure to facilitate gene transcription. Multiple epigenetic regulatory mechanisms are involved during early cardiac development, when progenitor cells differentiate into cardiomyocytes. Among these, the acetylation of histone H3K27 is particularly important [74]. Additionally, some lncRNAs can recruit chromatin-modifying enzymes such as H3K27 acetyltransferases to increase the acetylation levels of histones in the promoter regions of cardiac-related genes, thereby maintaining a more open chromatin state and promoting the efficient expression of key genes required for proper cardiac development [75].

## 4. Mechanism of Action Super-Enhancer RNAs

seRNAs modulate gene expression through multiple mechanisms, including recruiting various TFs and cofactors near the transcription start site, promoting the formation and stabilization of the transcription initiation complex, thereby regulating the expression of associated genes. Additionally, seRNAs interact with transcription elongation factors such as RNA polymerase II to promote the continuity and efficiency of transcription. seRNAs can also modify chromatin structure and regulate histone modifications, rendering the chromatin more accessible for TF binding (Figure 3). Chromatin remodeling enhances gene transcription and is involved in transcriptional modifications [76]. p300/CBP (CREB-binding protein), a critical transcriptional co-activator, plays a key role in cell growth, differentiation, and development and is essential for cell cycle progression. seRNAs recruit CBP to active enhancer regions, where they interact with the histone acetyltransferase domain of CBP, thereby promoting CBP acetylation and transcription of related target genes [77].

A widely recognized and significant mechanism of action of seRNA is the formation of stable enhancer-promoter (E-P) chromatin loops. Studies have shown that seRNAs recruit cohesin proteins to enhance the dynamic stability of the E-P loop, thereby facilitating mRNA transcription [78] (Figure 3). Additionally, seRNAs are involved in the formation of local chromatin R-loop structures, promoting long-range interactions between SEs and gene promoters and altering the higher-order genomic structure to regulate target gene expression [79]. However, because of the variety in spatiotemporal expression and structural heterogeneity of eRNAs, their specific functional mechanisms in the pathophysiology of CVDs remain to be further explored.

## 5. The Role of Super-Enhancer RNAs in Cardiovascular Diseases

Recent studies have highlighted the crucial roles of specific seRNAs in the development of CVDs, including atherosclerosis [80], myocardial infarction (MI) [38], heart failure [49], and disruptions of vascular homeostasis [81] (Figure 4; Table 2). seRNAs can affect the normal physiological functions of the cardiovascular system by regulating gene expression at multiple levels. SE-associated RNAs play essential regulatory roles in various cardiovascular cell types, including vascular smooth muscle cells (VSMCs), endothelial cells, cardiomyocytes, and immune cells. They are involved in key biological processes such as cell proliferation, migration, differentiation, apoptosis, inflammation, and lipid metabolism. In a healthy cardiovascular system, seRNAs exhibit distinct expression patterns in cardiomyocytes, endothelial cells, and smooth muscle cells (SMCs). Cardiomyocytes are involved in regulating the normal expression of genes related to cardiac contraction and maintaining proper heart contraction function. Endothelial cells regulate the expression of genes associated with vasodilation and angiogenesis, thereby ensuring proper physiological function of blood vessels.

An in-depth exploration of seRNA mechanisms is anticipated to provide new insights into the pathophysiology of CVDs, potentially leading to the development of novel diagnostic tools and therapeutic strategies. Such advancements could not only improve treatment efficacy but also enhance patient prognosis by offering more targeted and effective interventions.

### 5.1. The Role of Super-Enhancer RNAs in Cardiac Development and Repair

Recent studies have revealed the crucial role of lncRNAs associated with cardiac SEs in the regulation of cardiac development and repair [38,49,82]. One such lncRNA, cardiac mesoderm enhancer-associated non-coding RNA (CARMEN, also referred to as CARMN), was first identified in fetal hearts and has since been shown to play a key role in cardiac cell lineage specialization and differentiation [49]. Specifically, CARMEN expression was significantly upregulated at the onset of cardiac differentiation, promoting the expression of cardiac cell lineage-specific genes. CARMEN knockdown significantly inhibits differentiation of cardiac precursor cells (CPCs).

Further investigations have shown that CPCs isolated from adult hearts predominantly differentiate into SMCs but can be redirected to cardiomyocyte fate through transient modulation of the NOTCH signaling pathway [83]. CARMEN, which is transcribed from an enhancer region in the miR-143/145 locus, responds to NOTCH signaling and is essential for this process [84]. While the transcription of CARMEN is largely independent of miR-143/145 [85], its expression is tightly linked to NOTCH activity. Inhibition of NOTCH signaling results in the downregulation of CARMEN, which in turn suppresses miR-143/145 expression and inhibits SMC differentiation. This suggests that the CARMEN/miR-143/145 axis plays a crucial role in CPC differentiation, making it a promising target for promoting cardiomyocyte differentiation. Therefore, CARMEN serves as a key effector of the NOTCH signaling pathway and is a critical regulator of cardiac cell fate determination [84].

In addition to CARMEN, another SE-associated lncRNA, Mhrt (my heart), transcribed from the Myh7 SE region, plays a vital protective role in the cardiac stress response and remodeling [86]. Mhrt is highly expressed in the adult heart, where it protects against cardiomyopathy by preventing the activation of cardiac gene expression in response to pathological stress. Specifically, Mhrt competes with Brg1 to prevent the activation of cardiogenic genes that contribute to pathological cardiac remodeling. Mhrt expression is downregulated in hypertrophic, ischemic, or idiopathic cardiomyopathy and heart failure, and restoring its expression has been shown to improve cardiac function, prevent hypertrophy, and inhibit progression to heart failure.

Additionally, researchers have identified a cardiac-specific circular RNA, circNfix, which is SE-regulated and plays an important role in cardiac regenerative repair [38]. It is highly expressed in human, rat, and mouse hearts and functions through two main mechanisms. First, circNfix enhances the interaction between Ybx1 (Y-box binding protein 1) and Nedd4l (E3 ubiquitin ligase), leading to Ybx1 degradation via ubiquitination. This suppresses the expression of cyclin A2 and B1, thereby inhibiting cardiomyocyte proliferation. Second, circNfix acts as a molecular sponge for miR-214, increasing the expression of GSK3β (a protein) and blocking β-catenin signaling, which is crucial for cardiomyocyte growth and angiogenesis [87]. This further suppresses heart cell proliferation and repair after myocardial infarction. circNfix downregulation promotes cardiomyocyte proliferation and angiogenesis while inhibiting cardiomyocyte apoptosis following myocardial infarction, thereby improving myocardial function and prognosis. Hence, circNfix may be a valuable therapeutic target for enhancing postinfarction heart repair.

The repair process after myocardial infarction is often accompanied by the reactivation of the fetal genetic program [88,89]. This process is thought to be driven, at least in part, by lncRNAs associated with SEs. Studies have shown that many cardiac-related enhancer lncRNAs play important roles in cardiac development and pathology [82,90]. For example, Novlnc6 is tightly linked to key cardiac developmental genes such as Bmp10 and Nkx2-5 [6]. Notably, Novlnc6 shows differential expression across various cardiovascular pathological conditions such as dilated cardiomyopathy and aortic stenosis [91], suggesting its potential as a marker or regulator of these diseases.

Further research identified several enhancer-associated lncRNAs that are strongly associated with cardiovascular cell identity genes. Researchers have hypothesized that many of the newly identified lncRNAs are SE-associated lncRNAs. More specifically, SE regions appear to preferentially encode multi-exon, polyadenylated, and unidirectionally transcribed lncRNAs, which may have functional relevance in maintaining cardiac cellular identity [82].

### 5.2. The Role of Super-Enhancer RNAs in Hypertrophy and Cardiomyopathy

Cardiomyopathies encompass a group of diseases that affect the myocardium and potentially lead to heart failure, arrhythmias, and other clinical manifestations. Several recent studies have suggested that seRNAs play key roles in myocardial damage repair and the development of cardiomyopathy.

A recent study identified Snhg7, a novel lncRNA driven by SEs, as a key player in cardiac hypertrophy [39]. Snhg7 induces ferroptosis (a form of iron-dependent cell death) and contributes to cardiac hypertrophy by interacting with T-box protein 5 (Tbx5), a critical TF for cardiac development and function [92]. This interaction is involved in the transcriptional regulation of key genes, including glutaminase 2 (GLS2), which is crucial for cellular metabolism and the hypertrophic response. In addition, the core TF Nkx2-5 directly binds to its own SE, lncRNASnhg7, resulting in increased activation of both. These findings suggest that modulation of seRNAs and their regulatory networks may be a novel strategy for treating cardiac hypertrophy.

Disturbed myocyte metabolism is a key factor in the development of diabetic cardiomyopathy [93]. It has been reported that the expression of peroxisome proliferator-activated receptor α (PPARα)-seRNA is significantly upregulated in cardiomyocytes under high glucose and palmitate stimulation. By disrupting glycolipid and energy metabolism in cardiomyocytes, PPARα-seRNA aggravates the pathological processes of diabetic cardiomyopathy, including cardiac dysfunction, myocardial fibrosis, and hypertrophy [94]. Additionally, seRNAs regulate adipose tissue metabolism. lncASIR, an adipocyte-specific seRNA, plays a critical role in regulating insulin response and maintaining adipocyte function. It modulates key metabolic pathways in adipocytes, including PPAR signaling, lipolysis, and adipocytokine signaling. Silencing of lncASIR leads to dysregulation of adipocyte metabolism, a process that may be closely related to metabolic dysregulation in CVDs, especially in the context of metabolic diseases such as obesity and insulin resistance [95].

### 5.3. The Role of Super-Enhancer RNAs in Heart Failure

seRNAs play important roles in the progression of heart failure. Wisper (Wisp2 SE-associated RNA) is a cardiac fibroblast-enriched lncRNA that plays a key role in cardiac fibrosis. Wisper is highly expressed in cardiac fibrosis following injuries such as MI and aortic stenosis. It exacerbates myocardial fibrosis and remodeling by regulating the proliferation, migration, and survival of cardiac fibroblasts, thereby contributing to the worsening of heart failure. Wisper also participates in collagen cross-linking and matrix stabilization by interacting with TIA1-related protein, enabling it to regulate the expression of the pro-fibrotic form of lysyl hydroxylase 2 (PLOD2). Moreover, antisense oligonucleotide-mediated silencing of Wisper attenuates the pathological development of MI-induced fibrosis, prevents adverse remodeling of the damaged heart, and improves cardiac dysfunction [21]. This suggests that Wisper is a promising therapeutic target for heart failure.

Another study screened and identified a novel seRNA, LINC00881, which is expressed in cardiomyocytes and regulated by GATA4-responsive SEs. It is abundantly expressed in the adult heart but is highly methylated and downregulated in heart failure. As an important regulator of calcium cycling in cardiomyocytes, LINC00881 affects cardiac function by regulating key calcium channels and myocardial niche organization genes, such as MYH6, CACNA1C, and RYR2 [96].

### 5.4. The Role of Super-Enhancer RNAs in Vascular Diseases

seRNAs have emerged as crucial regulators of vascular homeostasis and disease. For instance, LINC00607 expression is elevated in endothelial cells under high glucose and TNFα conditions, where it contributes to endothelial dysfunction. Inhibition of LINC00607 disrupts RNA-chromatin interactions, leading to reduced expression of SERPINE1, a key pro-inflammatory and pro-fibrotic gene, and diminished monocyte adhesion [97]. Further studies have shown that LINC00607 is highly expressed in arteries, especially under pathological conditions such as diabetes mellitus, and influences endothelial and SMC functions such as angiogenesis, expression of vascular endothelial growth factors, and cell proliferation. Under hyperglycemic conditions, silencing LINC00607 reverses extracellular matrix remodeling, inflammation, and fibrosis, indicating its therapeutic potential in vascular function regulation [81].

In vascular injury, CARMENs are significantly reduced in the cerebral arteries of human aneurysms, atherosclerotic arteries, and in mouse vascular disease models. Studies have suggested that CARMENs interact with myocardin (MYOCD), an activator of SMC-specific genes, to regulate the contractile phenotypes of VSMCs. The SMC-specific deletion of CARMENs significantly exacerbates intimal hyperplasia induced by vascular injury [85].

In calcific aortic valve disease, the SE-derived lncRNA LINC01013 is upregulated. It may spatially coordinate with cellular communication network factor 2 (CCN2). Increased expression of LINC01013 promotes CCN2 expression and collagen synthesis, which are involved in regulating the fibrotic response and vascular remodeling [98].

### 5.5. The Role of Super-Enhancer RNAs in Atherosclerosis

Atherosclerosis is a chronic inflammatory disease involving vascular endothelial cells, SMCs, macrophages, and inflammatory cells. Studies have shown that seRNAs influence the progression of atherosclerosis by modulating immune responses, inflammation, cell proliferation, apoptosis, and lipid metabolism.

A previous study found that nuclear factor kappa B (NF-κB) synergizes with the BET bromodomain to form SEs that drive the transcription of inflammatory genes and exacerbate inflammatory responses. Inhibition of NF-κB and BET bromodomain can attenuate SE-mediated inflammatory gene transcription and reduce inflammatory diseases such as atherosclerosis [99]. Further studies revealed that NF-κB and BET bromodomains form SEs in the upstream region of classical inflammation-associated microRNAs (miR-146a/155) to drive their transcription. In other words, SEs not only promote the expression of classical inflammatory genes but also regulate the transcription of inflammatory miRNAs. These SE-associated miRNAs, in turn, negatively regulate the inflammatory pathway by targeting inflammatory mediators, such as TRAF6, thereby maintaining an appropriate inflammatory response [100].

Recent research has shown that ABCA1-seRNA, which is located in intron 1 of ABCA1, plays an important role in high-density lipoprotein (HDL) biosynthesis [80]. ABCA1-seRNA knockdown inhibited ABCA1 expression and decreased APOA1- and HDL-mediated cellular cholesterol efflux, resulting in increased intracellular lipid accumulation. Mechanistically, ABCA1-seRNA binds specifically to the MED23 protein, forming an intermediary complex that recruits the TFs RXRa and LXRa to activate the ABCA1 promoter via RNA polymerase II. This enhances the transcriptional activity of ABCA1, which plays a crucial role in the maintenance of cholesterol homeostasis. Additionally, the absence of ABCA1-seRNA induces M1 polarization in macrophages and promotes cell proliferation, migration, and adhesion. Furthermore, ABCA1-seRNA promotes the ubiquitination of the TF P65, which inhibits the activation of the NF-κB signaling pathway and reduces the secretion of inflammatory cytokines, thereby exerting an anti-inflammatory effect. Therefore, ABCA1-seRNA, as a novel epigenetic regulator, provides important theoretical support for reducing the risk of CVD, preventing and controlling atherosclerosis-related diseases such as coronary heart disease, and lays the foundation for the development of lipid-lowering drug targets. Interestingly, a recent study highlighted the role of neurotensin (NT) in lipid metabolism and atherosclerosis progression, showing that NT accelerates atherosclerotic plaque formation by remodeling the plasma triglyceride pool, independent of changes in circulating cholesterol and triglycerides [101]. This suggests potential interactions between lipid metabolism pathways and epigenetic regulators like seRNAs, providing valuable insights for future research targeting lipid-related cardiovascular diseases.

Additionally, studies have reported that during Toll-like receptor 4 (TLR4) signaling, seRNAs dynamically induce the expression of key genes associated with innate immunity and inflammatory responses. Genes suppressed by TLR4 signaling are also linked to SE structural domains, accompanied by substantial suppression of eRNA transcription. Characterization of eRNA transcription patterns in SE regions reveals their regulatory roles in immune responses and inflammatory processes [25].

In addition to their involvement in inflammation and lipid metabolism-related pathways, another key feature of atherosclerosis is the migration and phenotypic transformation of SMCs. seRNAs may regulate SMC-associated genes and influence plaque formation and stability by promoting SMC proliferation and migration. The function of CARMEN in various CVDs has been increasingly revealed [85,102,103]. In atherosclerosis, the interactions between CARMEN and its associated miRNAs are highly complex, with both synergistic and independent mechanisms potentially at play [103]. In one study, researchers used GapmeRs to silence CARMEN in human coronary artery smooth muscle cells and employed CRISPR-Cas9 to achieve full gene knockout of CARMEN in an adeno-associated virus-PCSK9-induced mouse model. The results indicated that CARMEN deficiency promotes VSMC proliferation and migration, accelerates atherosclerosis progression, and exacerbates plaque formation [103]. In contrast, another study utilized GapmeR-mediated knockdown of CARMEN expression in the aortas of Ldlr-/- mice and observed that the suppression of VSMC proliferation and migration led to a significant reduction in atherosclerotic lesion formation [102]. These contrasting findings may stem from differences in experimental models, specific targeting domains of GapmeRs, and duration of atherosclerosis development [101]. Despite these discrepancies, the divergent results have provided important insights into the complex role of CARMEN in atherosclerosis. They also offer valuable perspectives for future experimental design and methodological considerations.

### 5.6. The Role of Super-Enhancer RNAs in Pulmonary Hypertension

seRNAs play crucial roles in the development of pulmonary hypertension. CircKrt4, an SE-associated circular RNA, is significantly upregulated in the pulmonary artery endothelial cells (PAECs) under hypoxic conditions. This seRNA promotes the pathological transformation of endothelial cells to mesenchymal cells through transcriptional activation of N-calmodulin in the nucleus and also inhibits the translocation of glycerol kinase to the mitochondria from the cytoplasm. This disruption contributes to oxidative stress and mitochondrial dysfunction, exacerbating endothelial cell dysfunction and accelerating the progression of pulmonary hypertension [20].

Another study found that a circular RNA derived from NAP1L4, circNAP1L4, was significantly downregulated in hypoxia-induced human pulmonary artery smooth muscle cells (HPASMCs). Overexpression of circNAP1L4 regulates glycolysis by modulating NAP1L4-mediated SE modification and affecting the expression of hexokinase II (HKII). This results in the inhibition of PASMSC proliferation and facilitates cell–cell communication between HPASMCs and HPAECs, ultimately alleviating the progression of hypoxia-induced pulmonary hypertension [104].
biomedicines-13-00117-t002_Table 2Table 2Functions and roles of seRNA in cardiovascular diseases.DisorderseRNAFunctionCardiac Development and RepairCARMENRegulates cardiac precursor cell differentiation and promotes heart-specific gene expression [49].MhrtProtects against pathological cardiac remodeling by inhibiting harmful gene activation [86].circNfixInhibits cardiomyocyte proliferation and repair after myocardial infarction by degrading Ybx1 and blocking β-catenin signaling [38].Novlnc6Maintaining cardiac cellular identity, linked to key cardiac developmental genes (e.g., Bmp10 and Nkx2-5) [82,90].Hypertrophy and CardiomyopathySnhg7Drives cardiac hypertrophy by inducing ferroptosis and interacting with Tbx5 [39].PPARα-seRNADisrupts glycolipid and energy metabolism in cardiomyocytes, aggravating the pathological processes of diabetic cardiomyopathy [94].MhrtRestoring Mhrt expression helps prevent pathological changes in cardiomyopathy [86].Heart FailureWisperExacerbates myocardial fibrosis and remodeling by regulating fibroblast proliferation, migration, and survival [21].LINC00881Affects cardiac function by regulating calcium channels and myocardial niche organization genes.MhrtRestoring Mhrt expression improves heart function and prevents heart failure progression [86].Vascular DiseasesLINC00607Regulates vascular homeostasis and disease by influencing endothelial dysfunction, inflammation, fibrosis, and cell proliferation under pathological conditions such as high glucose and TNFα [96].CARMENRegulate the contractile phenotypes of VSMCs by interacting with myocardin (MYOCD) [85].LINC01013Upregulated in calcific aortic valve disease, promoting the expression of CCN2 and collagen synthesis, thereby regulating fibrotic responses and vascular remodeling [98].AtherosclerosismiR-146a/155Regulate inflammatory pathways by targeting inflammatory mediators like TRAF6, maintaining a balanced inflammatory response [100].ABCA1-seRNARegulates cholesterol homeostasis by enhancing ABCA1 expression and HDL-mediated cholesterol efflux, exhibiting anti-inflammatory effects through the inhibition of the NF-κB signaling pathway [80].CARMENRegulates smooth muscle cell (SMC) proliferation and migration, influencing plaque formation and stability in atherosclerosis [102,103].Pulmonary HypertensionCircKrt4Accelerate pulmonary hypertension by inducing pathological endothelial-to-mesenchymal transition and mitochondrial dysfunction in pulmonary artery endothelial cells (PAECs) under hypoxia [20].circNAP1L4Regulates glycolysis and inhibits pulmonary artery smooth muscle cell (PASMC) proliferation in pulmonary hypertension while also modulating cell communication between PASMCs and PAECs [104].


## 6. The Potential of Super-Enhancer RNAs as Biomarkers for Cardiovascular Diseases

seRNAs have become a new research hotspot in CVD, opening new directions for the exploration of CVD biomarkers and providing the scientific community with research targets to discover earlier and more specific diagnostic indicators. Here, we analyzed the potential of seRNAs as biomarkers for CVD.

Firstly, seRNAs are expressed in specific tissues and cell types with high specificity and sensitivity. SEs are located near genes associated with specific cell or tissue functions, and their regulatory RNAs are typically highly expressed in specific cardiovascular cells. The specific expression profile of seRNAs can reflect the potential risk of CVD even before the presentation of clinical symptoms. Changes in the levels of seRNAs in the blood serum or other body fluids can be used as early markers of disease and help in early diagnosis and intervention, thus reducing the incidence and mortality of CVDs. Moreover, the expression of seRNAs is usually stage-specific, involves phenotypic transitions in gene regulation, and exhibits dynamic changes. This makes them useful for monitoring disease progression and predicting prognosis.

Second, seRNAs are closely associated with disease mechanisms. seRNAs participate in complex gene regulatory networks, and changes in their expression levels may be associated with multiple key signaling and gene regulatory pathways. Thus, seRNAs may be used not only as a marker of disease but also to provide important clues for understanding the pathogenesis of CVD, unlike other biomarkers that simply reflect disease status.

Third, seRNAs have unique advantages over traditional protein markers (e.g., high-sensitivity C-reactive protein and cardiac troponin) and can complement current biomarker systems. Combined detection with other markers can improve diagnostic accuracy and reliability, especially for early diagnosis and disease subtype identification.

Finally, the regulatory roles of seRNAs provide new targets for personalized therapies. Through the development of specific inhibitors or activators, targeted modulation of specific seRNAs can alter the expression of relevant genes, thereby modulating the pathological process of CVDs. Especially in the context of developing gene editing and RNA intervention technologies, seRNAs are expected to become an important component of precision medicine for CVDs with the advancement of technology and in-depth research.

## 7. Approaches and Strategies for Super-Enhancer RNAs

Small-molecule drugs targeting seRNAs are often characterized by high specificity and selectivity. These small-molecule compounds with therapeutic effects were designed to precisely bind to specific seRNA targets to modulate their functions. Their design requires an in-depth understanding of their structural features and mechanisms of action to ensure that the drugs can effectively interfere with or promote relevant biological processes. For example, some small-molecule compounds may interact with key bases or structural domains on the seRNA to change its conformation, thereby affecting its ability to bind to other molecules and ultimately modulate gene expression and cardiovascular physiological functions.

Recently, several studies have revealed that RNA interference (RNAi) and antisense oligonucleotides have broad application prospects in seRNA-based therapeutics. RNAi can specifically silence the seRNA expression, thereby inhibiting the regulation of genes related to CVD. Antisense oligonucleotides, on the other hand, can hybridize with seRNA and prevent it from performing its normal function. These nucleic acid drugs have the advantages of precise targeting and efficient regulation, which provide a new means for the treatment of CVDs. However, their application also faces challenges, such as drug stability, delivery efficiency, and potential immune response, which require further research and optimization. With advancements in technology, gene therapy is gradually being applied to complex diseases such as cancer and neurological disorders. This involves a series of planned and targeted methods and techniques aimed at patient-specific genes, with the goal of improving or restoring cellular function at the genetic level, ultimately achieving disease treatment.

Common gene editing technologies, such as CRISPR-Cas9, zinc finger nucleases (ZFNs), and transcription activator-like effector nucleases (TALENs), have their own advantages and disadvantages in regulating seRNA expression. The CRISPR-Cas9 system is easy to operate and has high efficiency but has potential off-target effects. ZFN technology is more specific but is relatively complicated to design and construct, and TALENs have higher precision under certain specific circumstances. When choosing a gene editing tool, it is necessary to comprehensively consider the editing efficiency, specificity, safety, and scope of application for precise regulation of seRNA expression.

## 8. Discussion

With the rapid development of high-throughput genomics in recent years, a large number of seRNAs have been discovered. Through the integration of transcriptomics, proteomics, and metabolomics, seRNAs were found to exhibit a wide range of biological functions. They play crucial roles in gene transcription as well as in the differentiation and development of cardiomyocytes and immune cells and are involved in the occurrence and progression of various immune-metabolic diseases, cancers, and CVDs. This review provides an in-depth discussion of the mechanisms and potential applications of seRNAs in CVDs.

In terms of CVD treatment, seRNAs have unique advantages as a potential drug target. Its powerful transcriptional regulatory ability and tissue specificity allow it to act more directly on key steps in cholesterol metabolism, thereby reducing side effects on other tissues. Furthermore, SE-derived RNAs, which are novel epigenetic regulatory factors, are reversible and controllable and offer more possibilities for drug development. However, research on seRNAs in CVDs is still in its early stages, and many issues remain to be explored. For instance, the specific regulatory networks of seRNAs are not yet fully understood, and their roles in different CVDs may vary. Additionally, the accuracy and reliability of seRNAs as risk prediction factors need to be further validated with more clinical samples and long-term follow-up studies.

Future research directions include the following: first, to further study the regulatory networks of seRNAs and clarify their specific mechanisms in CVDs; second, to conduct more clinical studies to validate the accuracy and clinical application value of seRNAs as risk prediction factors; and third, to develop cardiovascular drugs based on seRNAs, providing new strategies and methods for treating CVDs.

In conclusion, the role of seRNAs in CVDs has attracted increasing attention, and these research findings provide new ideas and directions for the prevention and treatment of CVDs. With continued advancements in research, seRNAs are believed to play a significant role in the diagnosis, treatment, and prevention of CVDs.

## Figures and Tables

**Figure 2 biomedicines-13-00117-f002:**
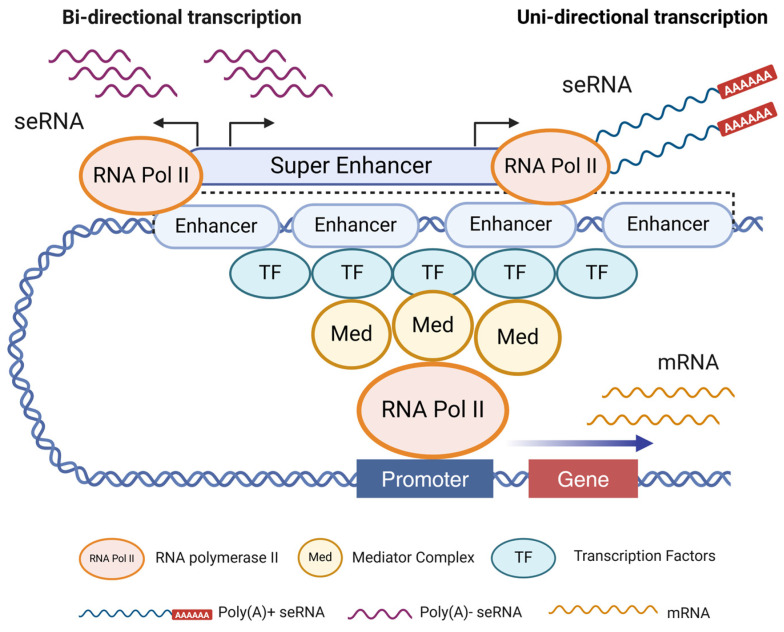
The biogenesis and classification of super-enhancer RNA. Super-enhancer RNAs (seRNAs) are non-coding RNAs transcribed from regions of the genome that contain super enhancers, which are clusters of enhancer elements driving the expression of key regulatory genes involved in cell identity and function. In bidirectional transcription, both sense and antisense seRNAs are produced simultaneously as short, non-polyadenylated RNAs. In unidirectional transcription, one type of long, polyadenylated, and capped RNA is generated (created in https://BioRender.com).

**Figure 3 biomedicines-13-00117-f003:**
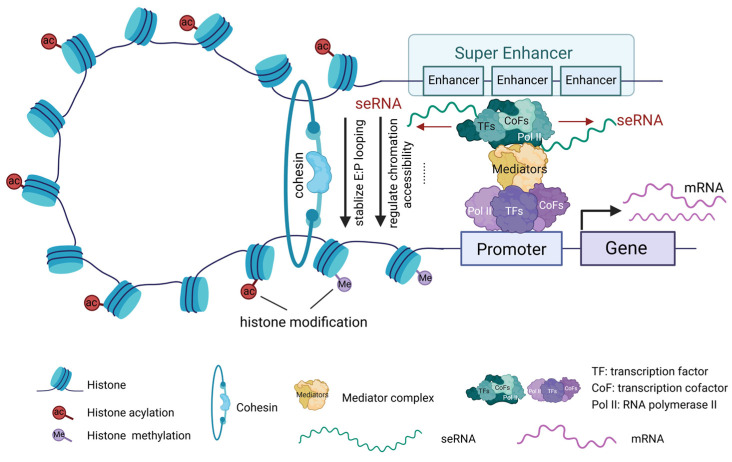
The action mechanisms of seRNAs. seRNAs regulate gene expression through various processes, such as modulating transcription, influencing chromatin structure, and interacting with other regulatory molecules (created in https://BioRender.com).

**Figure 4 biomedicines-13-00117-f004:**
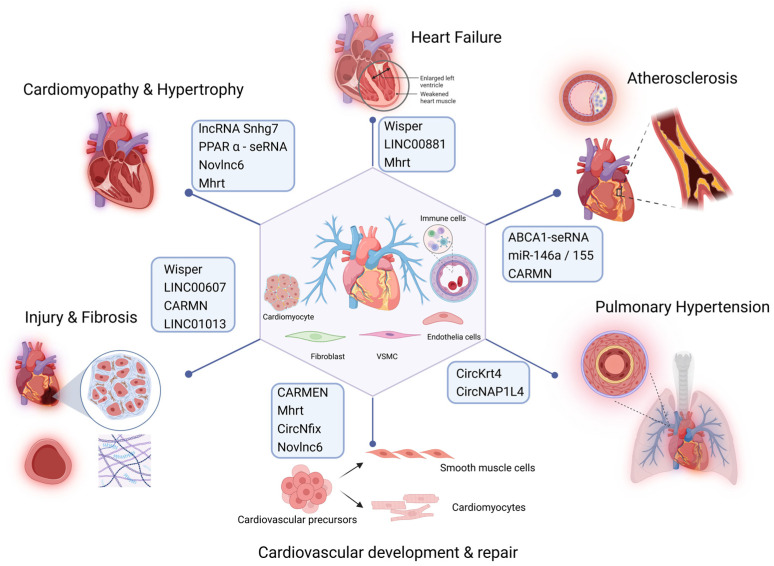
seRNAs in cardiovascular diseases. seRNAs play a key role in various cardiovascular diseases, including cardiomyopathy, atherosclerosis, myocardial infarction, heart failure, and vascular homeostasis by influencing biological processes like inflammation, cell differentiation, and tissue repair (created in https://BioRender.com).

## Data Availability

Not applicable.

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
