# Peer review of "Application Strategies of Super-Enhancer RNA in Cardiovascular Diseases"

_biomedicines, 2025, doi:10.3390/biomedicines13010117_

Round 1
Reviewer 1 Report
Comments and Suggestions for Authors
The manuscript entitled "Application Strategies of Super-Enhancer RNA in the Treatment of Cardiovascular Diseases" is a good work and contain state of art information on the role of super enhancer RNAs in cardiovasclar diseases. However, I have some suggestions on the manuscript before further consideration.
1. The abstract contains limited information only: I suggest including details on the basic concept of se-RNA
2. Since Super enhancer RNA belongs to the class of non-coding RNAs, a description of non-coding RNAs may be highly recommended in the introduction.
3. A table on the non-coding RNAs classification (including se-RNA) may be appropriate
4. A Methodology section describing the data collection and analysis methods is to be included
5. Authors can emphasis the role of se-RNAs on lipid metabolism by including studies like https://doi.org/10.1016/j.atherosclerosis.2024.117479
6. Minor typographic changes are required in some areas.
Reviewer 2 Report
Comments and Suggestions for Authors
I would like to thank the authors for their manuscript. The manuscript explores super-enhancer RNAs in the context of cardiovascular diseases, highlighting their potential as novel biomarkers. This manuscript is a timely and well-organized review that addresses an emerging area of research in CVDs. With minor adjustments, it has the potential to make a significant contribution to the field.
Minor comments:
1. In the introduction consider elaborating briefly on how seRNAs were first identified in other diseases or contexts before being applied to CVDs, to provide a broader background.
2. Line 58, The technical terms (e.g., H3K27ac, BET family proteins) should be briefly defined or referenced to make the text clearer.
3. Add a short example or analogy to clarify the unique properties of SEs compared to typical enhancers.
4. Some technical terms and abbreviations may not be immediately familiar to all readers. Ensure definitions are included when terms are first introduced.
Reviewer 3 Report
Comments and Suggestions for Authors
This is a very interesting article about seRNAs in cardiovascular disorders. The authors have successfully synthesized the main literature findings in the area.
The article can be published, as long as minor changes are made, including:
- the authors should re-review the manuscript to avoid duplication information. e.g. lines 50-51 and 86-87.
- the title should better reflect the contents of the manuscript, as it currently does not deal only or mainly with the treatment of cardiovascular disorders.
- the text should also be reevaluated for grammar/syntax errors. See e.g. line 186: "An Recent studies". The form should be either singular, as there is only one reference, or plural, but in this case should be added more references. Or line 244: "Further Cardiomyopathies..."
